# How conflict shapes evolution in poeciliid fishes

Andrew I. Furness [1], Bart J.A. Pollux[2], Robert W. Meredith[3], Mark S. Springer[4] & David N. Reznick [4]

In live-bearing animal lineages, the evolution of the placenta is predicted to create an arena for genomic conflict during pregnancy, drive patterns of male sexual selection, and increase the rate of speciation. Here we test these predictions of the viviparity driven conflict hypothesis (VDCH) in live-bearing poecilid fishes, a group showing multiple independent origins of placentation and extreme variation in male sexually selected traits. As predicted, male sexually selected traits are only gained in lineages that lack placentas; while there is little or no influence of male traits on the evolution of placentas. Both results are consistent with the mode of female provisioning governing the evolution of male attributes. Moreover, it is the presence of male sexually selected traits (pre-copulatory), rather than placentation (post-copulatory), that are associated with higher rates of speciation. These results highlight a causal interaction between female reproductive mode, male sexual selection and the rate of speciation, suggesting a role for conflict in shaping diverse aspects of organismal biology.

[1] Department of Biological and Marine Sciences, University of Hull, Cottingham Road, Hull HU6 7RX, UK. [2] Experimental Zoology Group, Wageningen University, 6708 WD Wageningen, The Netherlands. [3] Department of Biology, Montclair State University, Montclair, NJ 07043, USA. [4] Department of Biology, University of California, Riverside, CA 92521, USA. Correspondence and requests for materials should be addressed to A.I.F. (email: afurness001@gmail.com) or to D.N.R. (email: david.reznick@ucr.edu)

Genomic conflict leaves an imprint on genomic architecture, mode of inheritance, and diverse features of organismal biology[1,2]. The impact of conflict on the evolution of organismal biology changes in response to how organisms reproduce[3]. For sea urchins or abalone, which cast their gametes to the whims of ocean currents, conflict shapes the interface between sperm and eggs[4,5]. For organisms with internal fertilization, like birds or insects, conflict can be manifested in how a female chooses who to mate with and, if she mates with more than one male, how she can influence whose sperm fertilize her eggs[2]. Conflict shapes male morphology, behavior, and properties of the ejaculate, which can include evolving mechanisms that affect the ability of sperm to compete with one another or manipulate a female's inclination to mate again[2]. Here we explore how the manifestations of conflict change in concert with the evolution of how females provision their young in livebearing organisms that provide no postnatal care.

The females of many organisms fully provision eggs before they are fertilized, then either shed them to the external environment after fertilization or retain them inside themselves during development and give birth to live young. This type of provisioning is called lecithotrophy or yolk-feeding. In some species of livebearers, females continue to provision offspring after the egg is fertilized, which is given the general name of matrotrophy or mother-feeding. There are diverse mechanisms of matrotrophy, one of which is placentotrophy, defined as the integration of specialized embryonic and maternal tissues into a placenta, which sustains and provisions developing young[6]. This shift in provisioning embryos from before to after fertilization has evolved independently multiple times in most vertebrate classes[7] as well as in a diversity of invertebrate taxa[8].

Crespi and Semeniuk[9] proposed that the progression from egg laying to lecithotrophic livebearing to matrotrophic livebearing represents a progressive increase in the opportunity for embryos to influence maternal provisioning because of the prolonged and more intimate contact between mothers and developing young. The prolonged proximity of mother and developing offspring creates a portal for paternal influence, mediated through the paternal haploid genome of developing embryos, on maternal provisioning. The essence of the conflict, first articulated by Trivers[10], is that the quantity of resources that is best for the offspring to obtain from its mother is somewhat greater than is in the best interest of the mother to provide to the offspring. This inequality and the balance of costs and benefits borne by each party in the interaction (maximizing the number of offspring a mother can produce versus offspring maximizing their own fitness) is predicted to result in constant selection for increased maternal control over resource allocation versus increased embryonic control over resource acquisition. The diverse and bewilderingly complex interactions between mother and embryo associated with placentas are hypothesized to be manifestations of this conflict[9].

Zeh and Zeh[11] proposed the viviparity-driven conflict hypothesis (VDCH) to address the evolutionary consequences of the shift from egg laying to livebearing, but their logic applies well to the shift from lecithotrophy to placentotrophy. If a female makes her full investment in offspring before the egg is fertilized, her ability to influence who sires her offspring means being choosy about who she mates with. If lecithotrophic females mate with inferior males, they risk squandering their large pre-copulatory investment. Such choosiness can lead to the evolution of elaborate male morphology and courtship behavior associated with sexual selection[2]. In contrast, placental females produce tiny, inexpensive eggs[12,13]. They can reduce the risk of poor mate choice by mating with multiple males, enabling them to influence which sperm fertilize eggs or through post-zygotic

mechanisms of sexual selection, such as selective embryo abortion or the differential allocation of maternal resources to embryos based on genotype[9,11,14,15]. This shift pre to post-copulation in when mates are selected de-emphasizes the importance of choosing who to mate with and instead emphasizes the importance of mating with multiple males to access genetic diversity in the sperm gene pool. A consequence is that we should see less evidence of sexual selection on male morphology and behavior in species with placentotrophy.

The diverse consequences of conflict associated with how females provision eggs and embryos carries implications for the evolution of reproductive isolation and the rate of speciation[16]. Coyne and Orr[17] and Rice[18] proposed that sexual selection and sexual conflict could be engines of speciation because they accelerate the rate of evolution of pre-zygotic reproductive isolation. Investigators have evaluated this proposition in birds by asking whether lineages with sexual dichromatism (an index of sexual selection) have more species or higher rates of speciation[19–21]. The results are inconclusive. In contrast, there is strong evidence of a positive association between sexual selection and speciation rate in African lake cichlids[22], insects[23], and organisms that use bioluminescence as part of courtship, relative to sister clades that use bioluminescence for defense rather than courtship[24]. These statistical associations between organismal biology and speciation rate suggest that sexual selection and sexual conflict can accelerate the rate of speciation because of their impact on the evolution of pre-zygotic reproductive isolation.

Sexual conflict may also accelerate the rate of evolution of post-zygotic reproductive isolation as a by-product of the evolution of placentotrophy. Evidence for such acceleration is that mammals evolve post-zygotic reproductive isolation five to ten times faster than birds or amphibians[25,26]. Furthermore, the rate at which mammals evolve post-zygotic reproductive isolation depends on the structure of the placenta, possibly as a function of the scope of immunological interaction between mother and fetus[27]. Zeh and Zeh[11,28] predicted that the evolution of placentotrophy causes reproductive isolation among populations as a by-product of the different paths each population follows in resolving the persistent conflict between male and female genomes. A by-product of such different paths is that crosses between populations may fail to resolve conflicts, causing reductions in offspring viability. Inter-specific crosses among species of mammals and among populations of a placental species of fish in the family Poeciliidae support this prediction[29–31]. Zeh and Zeh further predict that this acceleration of the evolution of post-zygotic reproductive isolation will amplify the importance of post-zygotic reproductive isolation in the speciation process[11].

The fish subfamily Poeciliinae (sensu Parenti[32]), which includes popular pet-store fish like guppies, mollies, and swordtails, provides suitable material for testing the VDCH and the potential interactions between the evolution of maternal provisioning, sexual selection, and speciation rate. All species of Poeciliinae, save one, bear live young. Most species are lecithotrophic, but some are placentotrophic[33]. We quantify maternal provisioning with the "Matrotrophy Index" (MI), which is the estimated dry mass of the young at birth divided by the dry mass of the egg at fertilization. Most species have MI values <1, implying little or no maternal provisioning after the egg is fertilized because the embryo loses mass over the course of development. Some species have MI values as high as 120, meaning that young at birth weigh 120 times more than the egg at fertilization[13]. Poeciliinae also vary in the development of secondary sexual traits. Most species have males that look very much like females, which lack ornamentation and have cryptic coloration. Some species have males with conspicuous and often polymorphic color patterns, elaborate courtship displays, and ornaments like enlarged dorsal fins, sword-like

extensions of the caudal fin, or laterally compressed bodies. This combined variation in male and female traits creates the opportunity to assess the predictions of the VDCH and further to evaluate the relative contributions of pre- and post-copulatory reproductive isolation on the rate of speciation.

We have previously shown that the VDCH accurately predicts an association between male sexually selected traits and maternal provisioning. We quantified five indices of male signaling and behavior and found males of placental species are significantly less likely to: (1) be brightly colored, (2) have ornaments, or (3) engage in courtship displays[34]. These species also have more pronounced sexual size dimorphism (males smaller than females) and significantly longer gonopodia (metamorphosed anal fins used as the intromittent organ)[34]. In Poeciliinae, all of these changes are associated with a shift away from pre-copulatory mate choice toward covert male mating behavior and forced copulation[34–36].

Here we extend our inquiry to ancestral state reconstructions of male and female traits and analyses of how the evolution of either male or female traits affects the subsequent evolution of traits in the opposite sex. The most likely ancestral state of females was to lack superfetation and placentotrophy and of males was to lack all traits associated with pre-copulatory mate choice. We predict that the state of maternal provisioning will determine how male secondary sexual traits evolve[11]. An alternative hypothesis is that male attributes shape the subsequent evolution of female provisioning[37]. We show that the evolution of derived male traits

associated with pre-copulatory sexual selection only occurred in lineages that lack placentation. Placentas evolved in lineages that either did or did not have males with enhanced traits associated with pre-copulatory mate choice, but if such traits were present, they tended to be lost. We thus support Zeh and Zeh's[11] prediction that that mode of maternal provisioning influences the evolution of male traits. Lastly, we address whether there is an association between the evolution of these traits and the rate of speciation. If pre-copulatory reproductive isolation is the primary determinant of the rate of speciation, then lineages with more highly developed traits associated with sexual selection should also have higher speciation rates. If post-copulatory reproductive isolation dominates, then we should instead see higher rates of speciation in placental lineages. We find no association between placentation and speciation rate, but male sexually selected traits are associated with accelerated speciation. This result is consistent with pre-zygotic, rather than post-zygotic, reproductive isolation governing speciation rate in these fishes.

## Results

**Ancestral state reconstruction of female reproductive mode.** The common ancestor of the subfamily had internal fertilization, probably bore live young, probably lacked placentotrophy, and lacked superfetation (Figs. 1 and 2, Supplementary Figs. 1–4, Supplementary Tables 1 and 2). Superfetation is the capacity of

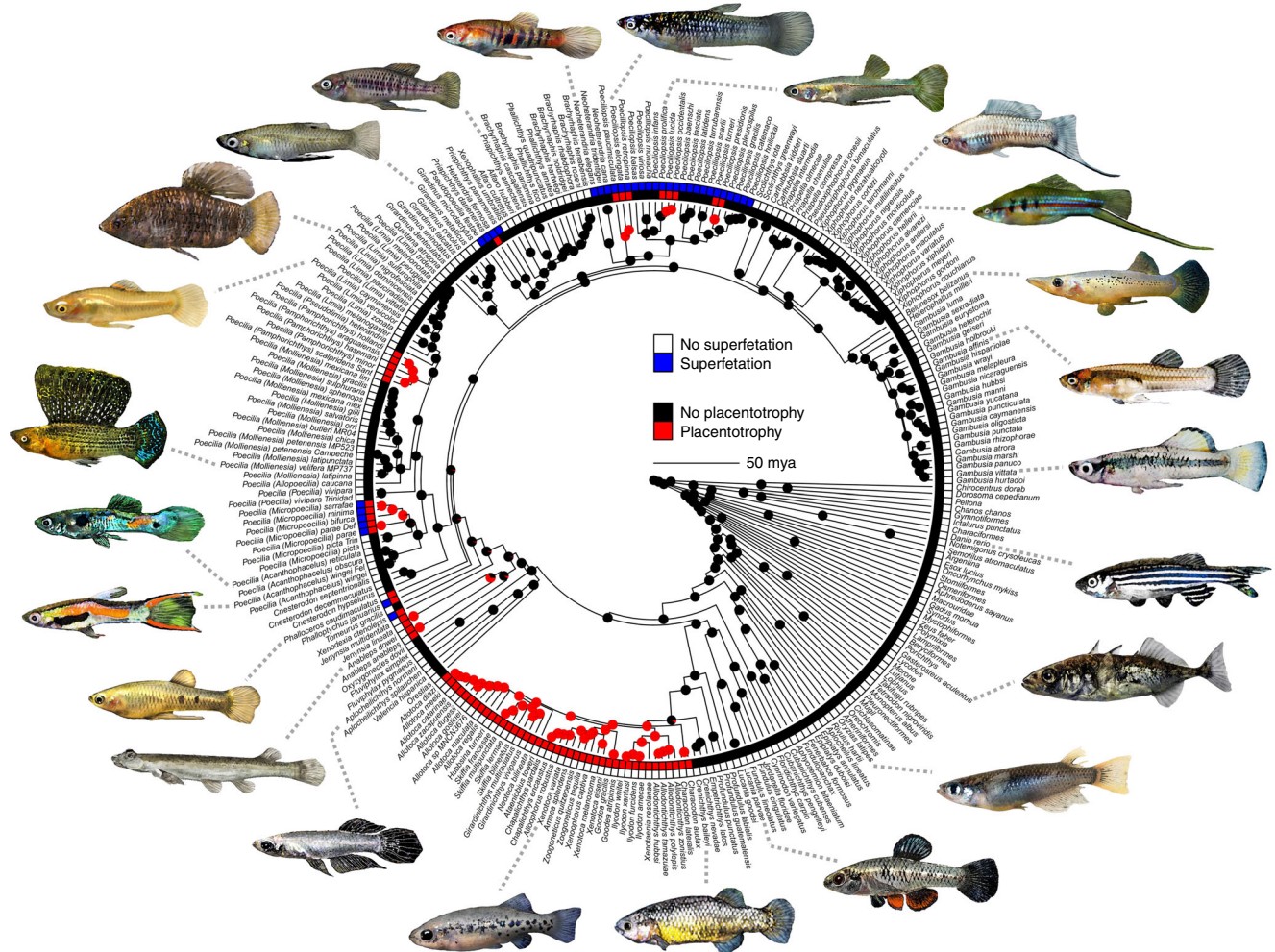

**Fig. 1** Maximum likelihood ancestral state reconstruction of placentotrophy. Circles at internal nodes indicate the likelihood (presence/absence) of placentotrophy estimated using the best-fitting equal rates model. The presence or absence of superfetation is also indicated on the tips of the phylogeny

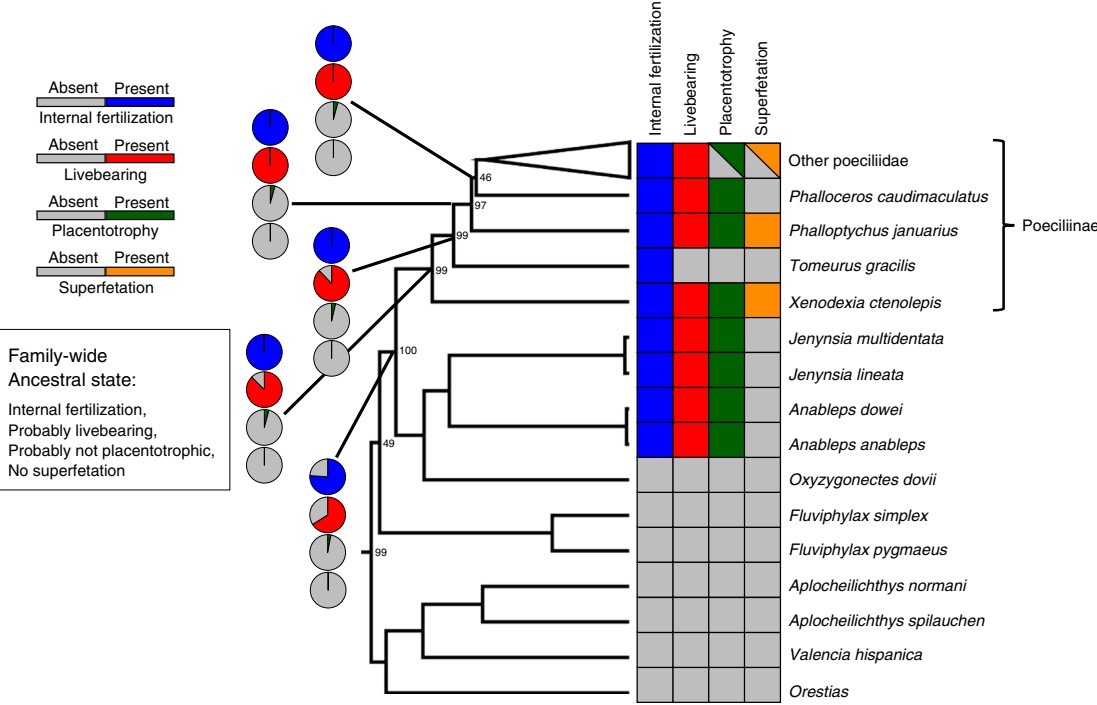

**Fig. 2** Ancestral reproductive mode of the subfamily Poeciliinae inferred using maximum likelihood. Circles at internal nodes indicate the likelihood (presence/absence) of each binary reproductive character estimated using the best-fitting equal rates model. Trees showing maximum likelihood ancestral state reconstructions on all ingroup and outgroup taxa can be found in Supplementary Figs. 1–4

females to carry multiple broods of young in different stages of development. Superfetation is known to promote polyandry in some mammals because sequential litters can be fathered by different males[38]. We predicted, and found, that the evolution of superfetation is associated with the evolution of placentation[34]. A consequence of superfetation in Poeciliinae is that females give birth more frequently and to fewer young per litter[13]. Livebearing could have been the ancestral trait of the subfamily, then lost in *Tomeurus*. Alternatively, livebearing evolved twice, once in *Xenodexia* and once in the common ancestor of the remainder of the subfamily, minus *Tomeurus* and *Xenodexia*. Maximum likelihood (ML) ancestral state reconstructions suggest that placentotrophy evolved nine times, and once gained, was never lost (Fig. 1, Supplementary Fig. 3; see also Supplementary Fig. 20 for ancestral state reconstruction using the continuous MI). The independent origins of the placenta are *Xenodexia*, *Phalloptychus*, *Phalloceros*, *Heterandria formosa*, the *Micropoecilia* and *Pamphorichthys* subclades of *Poecilia*, and three times within the genus *Poeciliopsis*. Superfetation evolved four times and was lost once. The loss was in the common ancestor of a clade of Central American species, including the genera *Brachyraphis*, *Phallichthys*, and *Alfaro* (Fig. 1). All possible orders of origin of superfetation and placentotrophy are present. Superfetation evolved before placentotrophy four times, placentotrophy and superfetation appear together three times, and placentotrophy evolved twice without the later evolution of superfetation (details in Fig. 1 and Supplementary Fig. 5). Placentotrophy and superfetation most often occur together. The exceptions are the genera *Phalloceros* and *Pamphorichthys*, which have placentotrophy but no superfetation, and *Priapichthys*, *Neoheterandria*, and some species of *Poeciliopsis*, all of which have superfetation but no placentotrophy.

**Ancestral state reconstruction of male attributes.** There is greater lability of male traits throughout the tree, which results in

less certainty in defining the properties of the common ancestor of the subfamily. ML analyses predict that the ancestral male most likely lacked courtship, sexual dichromatism, ornamentation, had a relatively long gonopodium, and was more likely to have mature males considerably smaller than females (Fig. 3, Supplementary Tables 3 and 4: Supplementary Figs. 6–10). Parsimony analyses reveal the same lability for internal nodes, yet predict that the ancestral state of males was to lack all attributes associated with sexual selection, meaning that courtship, sexual dichromatism, and ornamentation are all absent, plus males tend to have long gonopodia and are much smaller than females (Supplementary Table 4). On average, traits associated with sexual selection were absent in the common ancestor, then accumulate in some lineages. For subsequent analyses, we scored species as the number of dichotomous traits present (courtship present or absent, sexual dichromatism present or absent, and ornamentation present or absent), so that each species had a possible score of 0–3. The tree reveals the evolution of clusters of species with scores of 3 only in *Xiphophorus* and *Poecilia* (subgenera *Limia* and *Mollienesia*).

**Joint evolution of male and female traits.** The VDCH predicts that maternal provisioning will drive the evolution of male mating strategy and associated changes in morphology. Our evaluation of the interdependencies of the evolution of male traits associated with sexual selection and the evolution of maternal provisioning is simplified in the more likely ancestral state reconstruction in which placentotrophy is gained but never lost. If maternal provisioning governs the evolution of male traits, then male traits should be gained in lineages that lack placentotrophy and lost in lineages that have placentotrophy (blue arrows in Fig. 4). Conversely, if male traits influence the evolution of placentotrophy, then placentotrophy should be more likely to evolve in lineages that lack male traits associated with sexual selection (red arrows in Fig. 4).

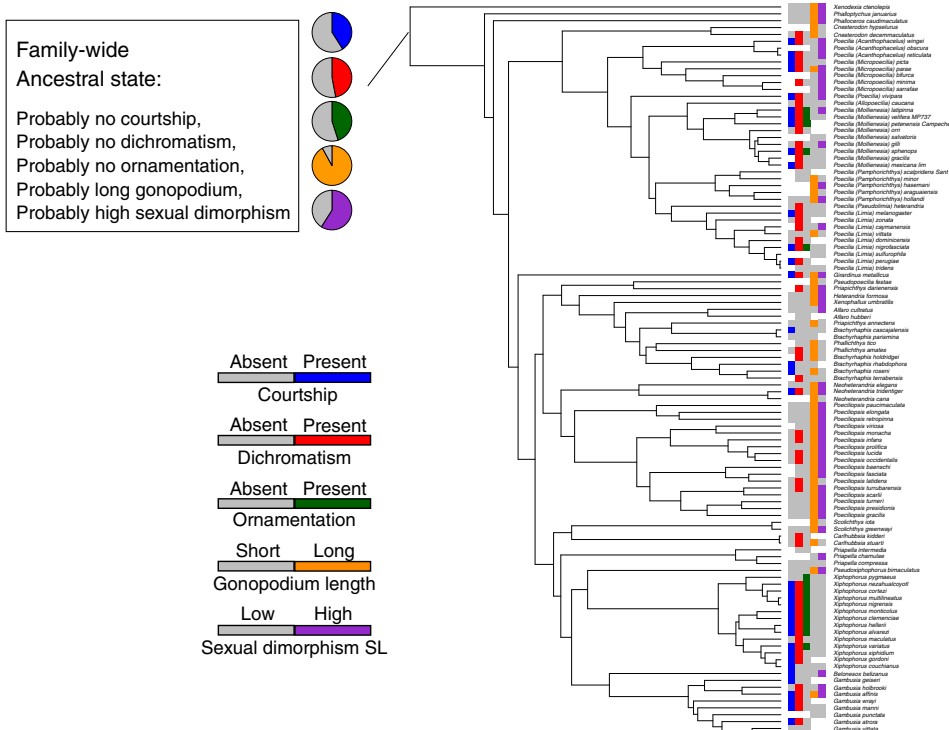

**Fig. 3** Ancestral male attributes of the subfamily Poeciliinae inferred using maximum likelihood. Circles at the root indicate the likelihood (presence/absence) of each binary trait estimated using the best-fitting model (Supplementary Table 3). Trees showing maximum likelihood ancestral state reconstructions on all internal nodes, for each trait, can be found in Supplementary Figs. 6–10

BayesTraits Discrete analyses reveal support for dependent (i.e., correlated) evolution between placentation and courtship, sexual dichromatism and sexual size dimorphism but not with ornamentation or relative gonopodium length (Supplementary Table 5; see also Supplementary Figs. 11–15 for a visualization of the joint evolution of male and female traits, Supplementary Tables 6–8 and Supplementary Figs. 16–18 for additional analyses that evaluate the robustness of the results reported here, and Supplementary Table 14 and Supplementary Fig. 21 for analyses that treat the MI as a continuous rather than as a dichotomous trait). The supported pathways in these analyses reveal that the evolution of all three male traits is conditioned on the mode of maternal provisioning, but the evolution of female traits being conditioned by male traits rarely attains strong statistical support (Fig. 4). There is strong support for courtship, sexual dichromatism, and sexual size dimorphism to be gained and lost in lineages that lack placentas (percentage of models in which the transition rate is 0, hereafter the $Z$-score, is 0–1% in 5 of the 6 cases) but very weak support for the evolution of placentation in lineages that have either courtship or sexual dichromatism ($Z = 80$ and 82%, respectively). The only instance of the evolution of maternal provisioning conditioned on male phenotype is the strong support for the evolution of placentas within lineages that have large sexual size dimorphism ($Z = 1$%), due to small male body size, which is the most likely ancestral trait of males. These results are thus most consistent with the mode of female provisioning driving the evolution of male attributes. An unexpected feature of these results is that male sexually selected traits are almost as likely to be lost as gained within non-placental lineages (discussed further below).

A more intuitive way to look at the same results is to simply enumerate the gains and losses of male and female traits as they appear on the tree. Traits associated with enhanced sexual selection in males (courtship, sexual dichromatism, and ornamentation) only evolve in lineages without placentation. Placentation can evolve in lineages with or without well-developed traits associated with sexual selection—six origins are in lineages that were predicted to have no male traits in the common ancestor. Three origins were in lineages that were predicted to have some sexually selected male traits in the common ancestor. Sexually selected male traits were subsequently lost in some species within two of these three lineages. We conclude that the weight of the evidence argues that how females provision their young shapes the evolution of male mating strategies.

**Speciation rate**. From the female perspective, Zeh and Zeh[11] predict that placentotrophic lineages will have a higher rate of speciation because of their higher rate of evolution of post-zygotic reproductive isolation[11]. In contrast, we found that lecithotrophic lineages have an estimated two-fold higher rate of speciation than placentotrophic lineages (Fig. 5, Supplementary Table 9, Supplementary Fig. 19; see also Supplementary Table 15 and Supplementary Fig. 22 for speciation rate analyses using the continuous MI). Zeh and Zeh's prediction is thus not supported. From the male perspective, courtship, sexual dichromatism, and ornamentation are each associated with significantly higher rates of speciation compared to lineages that lack these traits (Fig. 5, Supplementary Table 9, Supplementary Fig. 19). Speciation rate also increased as the number of dichotomous male traits associated with pre-copulatory mate choice (courtship, sexual dichromatism, ornamentation) increased. Lineages with all three traits present have significantly higher rates of speciation than those that have none of these traits (Fig. 6, Supplementary Table 9).

These analyses were performed with diversitree[39], which is known to be subject to false positive results[40]. We evaluated the

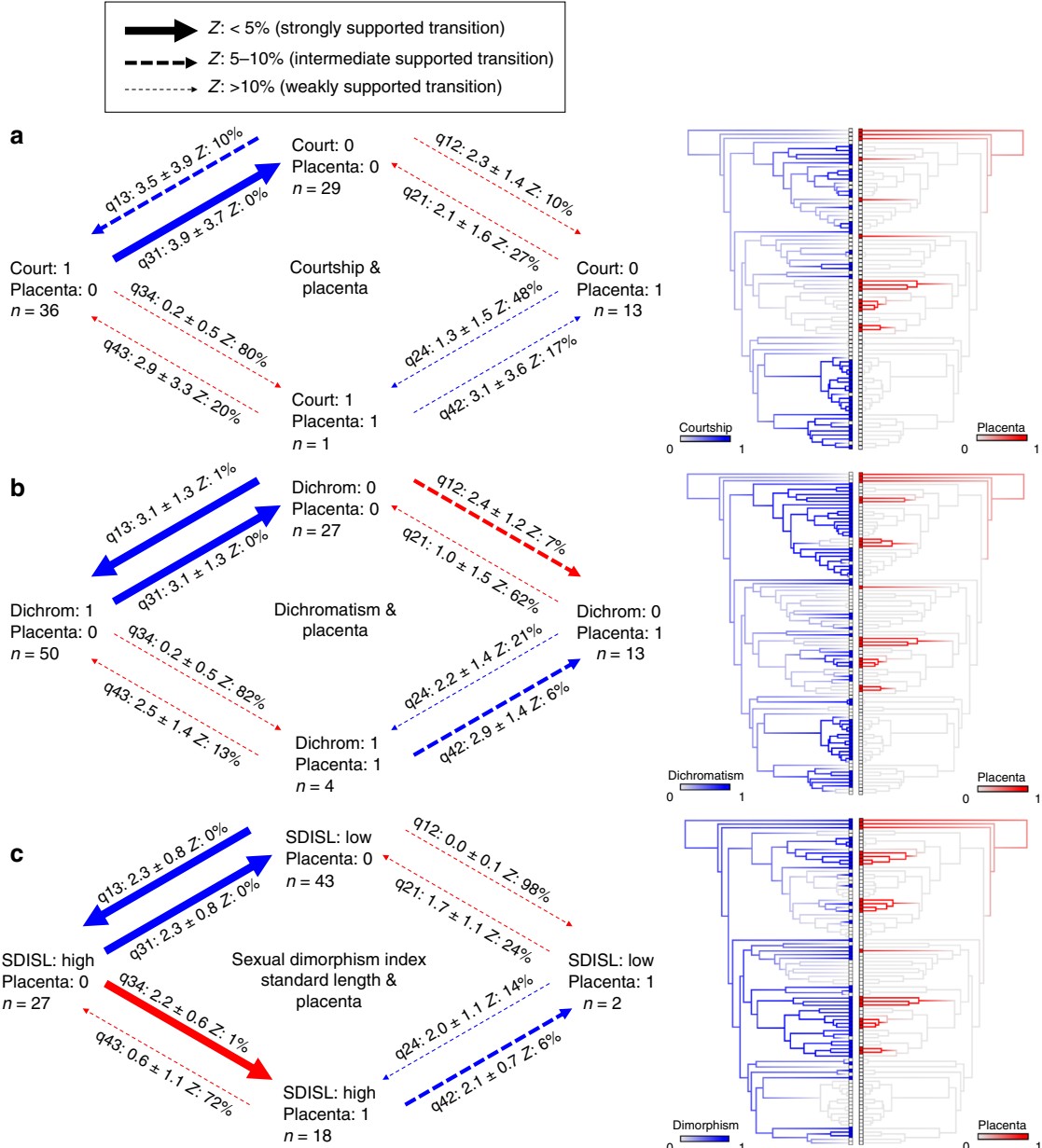

**Fig. 4** Correlated evolution of the placenta and male traits. Summary of transition rate estimates from BayesTraits Discrete dependent models of character evolution examining the joint evolution of the placenta and **a** courtship, **b** sexual dichromatism, and **c** sexual dimorphism index. Mean values, standard deviations, and the percentage of models in which each transition rate had a value of zero (Z) are summarized from the posterior distribution. Solid arrows indicate the best-supported evolutionary pathways in which Z-scores are <5%. Dashed arrows indicate less well-supported pathways, with arrow thickness scaled to reflect magnitude of transition rate (see key). Blue arrows indicate a change in male traits while female traits remain the same; conversely, red arrows indicate a change in female traits while male traits remain the same. Mirrored trees illustrate correlated evolution between male traits (blue) and the placenta (red). Ancestral state reconstructions by stochastic character mapping. Branch colors represent posterior probability densities of edge states based on 1000 stochastic character maps of each reconstruction

robustness of these results with two additional analyses, detailed in the supplemental materials, both of which yielded similar outcomes. First, we employed more conservative non-parametric tests (FiSSE[41]) for the three male binary traits and the mode of maternal provisioning (placentotrophy present or absent). The presence of courtship and ornamentation were each associated with a significantly higher rate of speciation. The presence of sexual dichromatism was associated with an increased rate of speciation but was not significant. The presence of placento-trophy was associated with a deceleration of the rate of speciation but also was not significant (Supplementary Table 10). Second, we

performed 100 simulations of the evolution of each binary trait using the actual tree and the estimated transition rates between the two character states. We estimated character-dependent diversification rate with diversitree[39] for all 100 trees, then used the distribution of these probabilities as a null distribution for evaluating the significance of the observed differences in diversification rate. By this criterion, the presence of each binary male trait was associated with a significantly higher rate of diversification than the absence of that trait (simulation test: $p <$ 0.05 in all three cases). The presence of placentotrophy was associated with slower diversification, but the difference was not

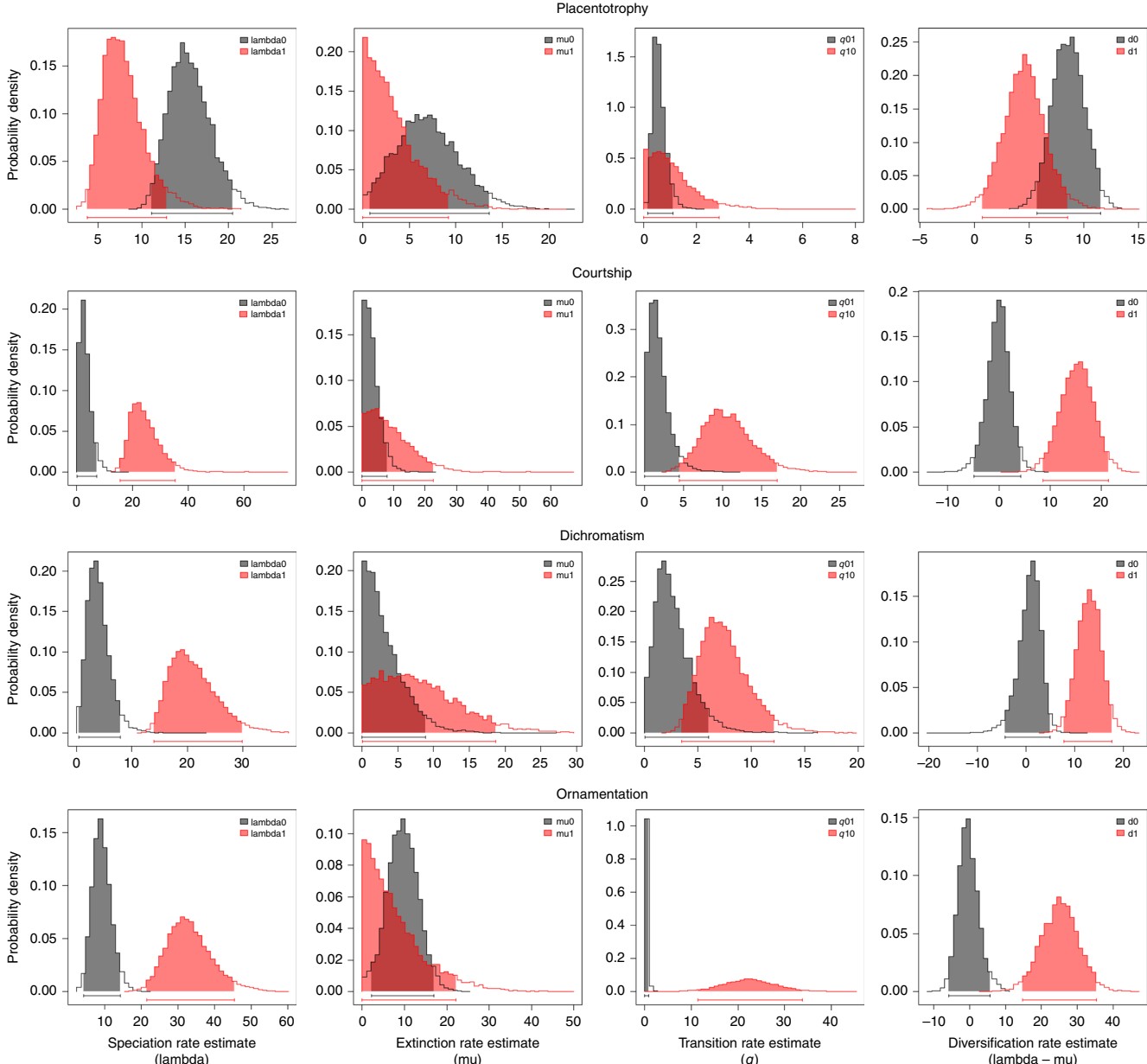

**Fig. 5** State-dependent diversification rates in Poeciliinae. Bayesian parameter estimates were inferred using the six-parameter binary state speciation and extinction model for each of the four binary traits—placentotrophy, courtship, sexual dichromatism, and ornamentation. Zero (0) indicates the absence of trait and one (1) indicates the presence. Estimates of trait-specific speciation rates (lambda); trait-specific extinction rates (mu); transition rates ($q$); and net diversification rates ($d$) calculated as the difference between speciation (lambda) and extinction (mu) rates. The 95% credibility intervals are indicated as horizontal colored bars above the $x$ axis

significant (simulation test: $p = 0.29$; Supplementary Table 11). These more conservative analyses thus provide robust support for the influence of male traits, but not the mode of maternal provisioning, on the rate of diversification.

A dilemma associated with these results is that both male and female attributes predict speciation rate and these attributes are in turn associated with one another. It is fair to ask whether we can conclude that male traits, independently of female mode of reproduction, predict the rate of speciation. The nature of the joint distribution of these traits makes such a separation possible because a near equal number of placental and non-placental lineages have males that lack all three dichotomous traits and some placental lineages have one or more of the dichotomous male traits (Supplementary Table 12). We addressed whether male or female attributes are the primary determinant of the rate

of speciation by creating models that contain both traits (Fig. 7, Supplementary Table 13). We constructed three such models with the "make.musse.multitrait" procedure in diversitree[39] representing all three combinations of the three male attributes associated with sexual selection and the mode of maternal provisioning (placenta present or absent). Figure 7 reports Bayesian Markov chain Monte Carlo (MCMC) posterior distributions of speciation "main effects" in the joint analyses of male and female traits. In all three analyses, the 95% credibility interval of the main effect of speciation rate for placentotrophic taxa (i.e., lambdaP) broadly overlaps 0, which means that the mode of maternal provisioning does not contribute significantly to the rate of speciation. In contrast, in all three analyses the main effect of speciation rate for taxa with courtship, sexual dichromatism, or ornamentation has a 95% credibility interval that is positive and does not overlap 0,

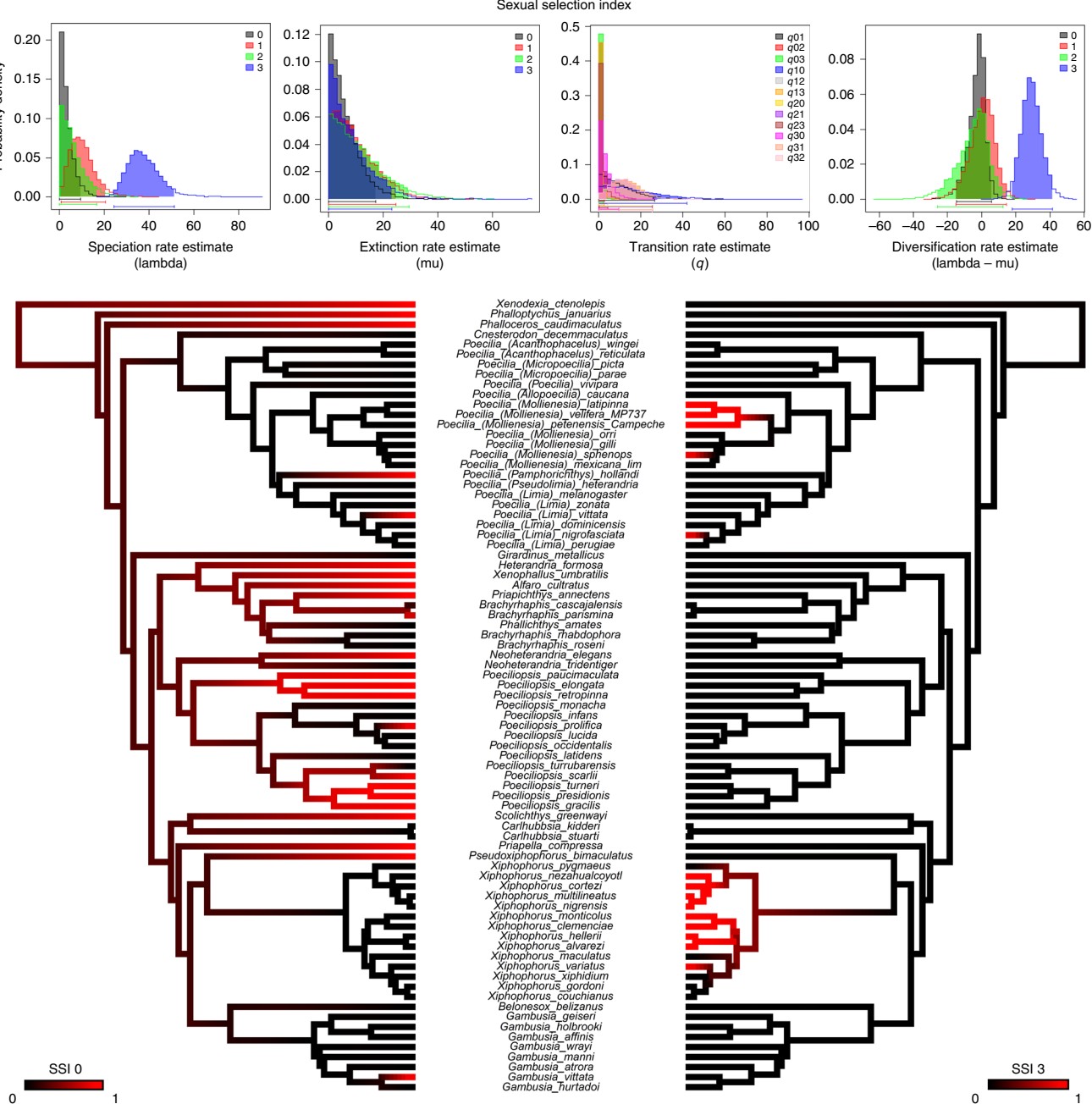

**Fig. 6** The relationship between the sexual selection index and diversification rate. Bayesian parameter estimates were inferred using the 20-parameter multiple state speciation and extinction model for the sexual selection index—a multi-state character with values ranging from 0, corresponding to the absence of courtship, sexual dichromatism, and ornamentation, to 3, indicating the presence of all 3 traits. Estimates of trait-specific speciation rates (lambda); trait-specific extinction rates (mu); transition rates (q); and net diversification rates calculated as the difference between speciation (lambda) and extinction (mu) rates. The 95% credibility intervals are indicated as horizontal colored bars above the x axis. Mirrored trees illustrating the contrast in branch length between species with a sexual selection index of 0 (absence of courtship, dichromatism, and ornamentation) versus a sexual selection index of 3 (presence of all 3 traits). The mean branch lengths for species with a sexual selection index of 0 is 20.6 ± 14.8 mya, while that of species with a sexual selection index of 3 is 2.7 ± 1.8 mya (mean ± sd). Ancestral state reconstructions by stochastic character mapping. Phylogeny restricted to species scored for the absence/presence of courtship, dichromatism, and ornamentation (n = 79). Branch colors represent posterior probability densities of edge states based on 1000 stochastic character maps of each reconstruction

suggesting that sexually selected male traits are associated with accelerated speciation rates. Taken together, these results support the argument that male traits alone determine the rate of speciation and the rate of speciation accelerates as males acquire traits associated with sexual selection (Fig. 7). These results in turn suggest that the evolution of pre-copulatory reproductive isolation plays a more important role than post-copulatory

reproductive isolation in causing the origin of new species in Poeciliinae.

## Discussion
Our results characterize how the evolution of female reproductive mode influences the evolution of male sexually selected traits,

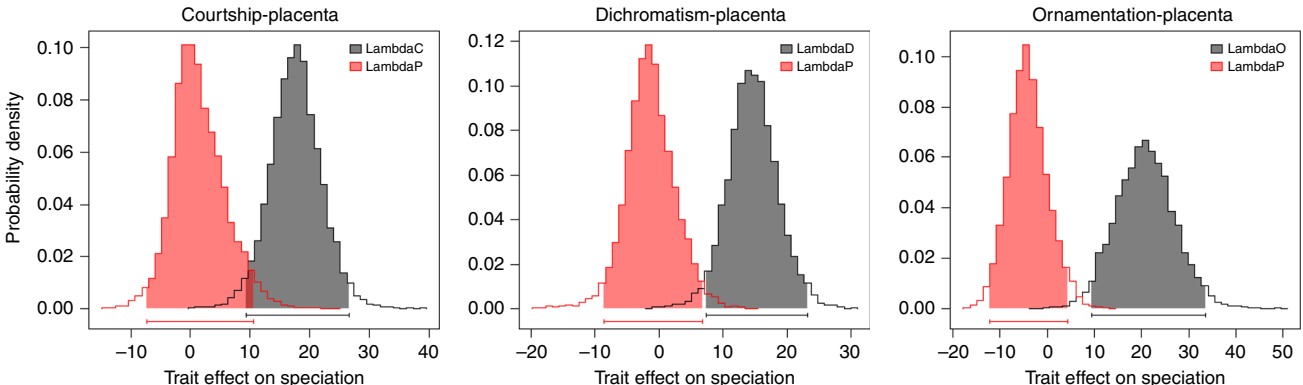

**Fig. 7** Joint analyses on the effect of male sexually selected traits and female placentotrophy on speciation rate. Bayesian Markov chain Monte Carlo posterior distributions of speciation "main effects" are presented from joint analyses of male and female traits. Ten-parameter models, which include main effects of the two traits on speciation and extinction and an independent model of character evolution (i.e., four transition rate parameters, gain and loss of each trait), were implemented in diversitree using the "make.musse.multitrait" command. These analyses were used to investigate whether male sexually selected traits or placentation was a stronger predictor of speciation rate. lambdaC/D/O refers to the main effect of speciation rate for taxa with courtship, sexual dichromatism, or ornamentation, respectively, and lambdaP refers to the main effect of speciation rate for placentotrophic taxa. We analyzed all paired combinations of male and female traits. The 95% credibility intervals for the effect of each trait on the rate of speciation are indicated as horizontal colored bars above the x axis. Male traits are gray and female traits are orange. The distribution of parameter estimates strongly suggests that male sexually selected traits are associated with a higher speciation rate and female placentation is not

which in turn affects the rate of speciation. Male traits associated with sexual selection never evolve in lineages with placentas. If placentas evolve in lineages that previously had sexually selected male traits, then the male traits tend to be lost. In the absence of placentas, sexual conflict can ignite the evolution of exaggerated male traits associated with sexual selection, causing accelerated speciation.

BayesTraits Discrete analyses strengthen our earlier support of the VDCH by establishing the order of appearance of male and female traits and showing stronger support for the evolution of male traits conditioned on female mode of reproduction, rather than the reverse. We had predicted that not only the rate of evolution of sexually selected traits would be higher in non-placental lineages but also that this change would be unidirectional, with such traits being gained at a higher rate. We instead found that male traits associated with sexual selection are almost as likely to be lost as gained within non-placental lineages. Similar losses of sexually selected traits have been documented in a diversity of organisms[42]. A general explanation for such reversibility is that the traits associated with sexual selection may also be associated with antagonistic natural selection. If so, then any shift in the balance between natural and sexual selection can cause either the elaboration or loss of such traits. Examples of specific causes might be selection against sexually selected traits, as by a predator, a loss of female preference, changes in the signaling environment that influence the conspicuousness of the trait, or as a by-product of the evolution of traits like male parental care.

Such losses have been reported in previous case studies of species of Poeciliinae. For example, *Poecilia latipunctata* has apparently lost sailfin ornamentation but retained courtship[43,44]. *Xiphophorus continens* has lost the caudal sword and vertical bars. It has retained degraded elements of courtship behavior but not courtship[45]. More generally, there is considerable diversity in the extent to which sexually selected traits are displayed in lecithotrophic species. This pattern argues that lecithotrophy creates a more permissive environment for the evolution of exaggerated sexual selection compared to placentotrophy, but other factors contribute to how these traits evolve and can select against them. Some of these other factors, such as predation or paternal care, reflect the mix of costs and benefits associated with the evolution of sexually selected traits. Others, like the evolution of female

preference, suggest a need to broaden our exploration of the factors that shape sexual conflict.

Rowe et al.[46] generalized models of sexual selection in ways that offer possible explanations for how conflict can cause both the loss and gain of sexually selected traits. Their models allow for the evolution of the threshold and sensitivity of female response to male traits. The reason to add such complexity to models is that mating biases can also be subject to natural selection[47], so the balance of natural versus sexual selection can be reflected in the evolution of both male traits and female preferences. With this added complexity, it becomes possible for sexual conflict to cause either the elaboration or loss of sexually selected traits in males, rather than simply igniting runaway sexual selection. Our observation of both the gain and loss of sexually selected traits within lecithotrophic lineages may well reflect such a dynamic balance between sexual and natural selection.

Placentation has been associated with accelerated evolution of post-zygotic reproductive isolation in broad comparisons among different classes of organisms[25,26]. Schrader, Travis, and Fuller[30,48] found substantial post-zygotic reproductive isolation in crosses among populations of *H. formosa*, a placentotrophic species in Poeciliinae; however, the absence of an association between placentotrophy and speciation rates in our results (Fig. 7) is consistent with the absence of any association between the evolution of post-zygotic reproductive isolation and speciation rate in reviews of the literature[49]. It is also consistent with the inference that post-zygotic reproductive isolation evolves after the fact of speciation, as a by-product of the long-term separation of species' gene pools[17], and with the observation that the rate at which post-zygotic reproductive isolation evolves is decoupled from the rate of speciation measured on geological timescales[50].

The elaboration of sexually selected traits, which are associated with pre-zygotic reproductive isolation, is strongly associated with higher rates of speciation in our data. This result suggests that pre-zygotic reproductive isolation can indeed play a significant role in the formation of new species. Similar associations between sexual selection and speciation are seen in a diversity of other organisms (e.g., cichlids[22,51], centrarchid fishes[52], insects[23], and organisms that use bioluminescence in courtship[24]). There are also associations between pre-zygotic reproductive isolation and speciation in plants (e.g., flowering time[53] or plant–pollinator

associations[54]). Our results thus add to those that support the importance of pre-zygotic reproductive isolation in driving speciation. What is new is that, in this particular subfamily of fish, there is an interaction between the evolution of the traits that appear to drive speciation rate and the evolution of how females provision their offspring. This interaction in turn reveals how the shift from pre- to post-fertilization provisioning of offspring shifts the venue of conflict in a way that suppresses the evolution of male traits associated with sexual selection and in turn results in a persistently slower rate of speciation. The converse is that the absence of post-fertilization provisioning is permissive to the evolution of elaborate male traits associated with sexual selection and, as a by-product, facilitates accelerated speciation.

## Methods

**Data set**. Taxa were scored for presence versus absence (0/1) of internal fertilization, viviparity, matrotrophy, and superfetation. These data were gathered for this study, complemented with reports in the literature. The outgroup taxa Atheriniformes, Characiformes, Ophidiiformes, and Scorpaeniformes are polymorphic for some reproductive mode characters; for ancestral state reconstructions, these taxa were excluded. For the subfamily Poeciliinae, the extent of pre-versus post-fertilization maternal provisioning of embryos has been quantified using the MI—defined as the ratio of the estimated dry mass of offspring at birth divided by the dry mass of an egg at fertilization[34]. Here we converted this continuous variable into a binary presence versus absence character (i.e., lecithotrophy versus placentotrophy). In doing so, we took a conservative approach so as to avoid false positives. Species were scored as having placentotrophy if the data were trustworthy (number of broods used to calculate MI is >7, and the range of embryo stages present in the collection is >25) and (1) the slope of the regression line that relates stage of embryo development to log-transformed embryo dry mass was significantly greater ($t$ test: $p < 0.05$) than 0 in at least one population, or (2) if two or more populations of a given species had an MI > 1, and the slope of the regression line that relates stage of embryo development to log-transformed embryo dry mass was significantly greater ($t$ test: $p < 0.05$) than −0.0071, which corresponds to an MI of 0.7. In particular, we scrutinized species with a reported MI between 0.7 and 2.0, indicating a slight degree of placentotrophy (these included some species in genera such as *Xiphophorus* and *Gambusia* traditionally considered lecithotrophic, as well as some species in the genus *Poeciliopsis*, which is known to contain both lecithotrophic and placentotrophic species). Based upon the stringent criteria detailed above, the majority of these species were scored as lecithotrophic. Data on presence or absence of superfetation, the ability to gestate multiple temporally overlapping litters fertilized at different time points, was obtained from Pollux et al.[34], with data for additional species obtained from the literature.

For species in the subfamily Poeciliinae, data on the presence versus absence (0/1) of courtship behavior (courtship), sexually dimorphic coloration (dichromatism), and exaggerated male display traits (ornamentation) was obtained from Pollux et al.[34]. The sexual selection index is the total number of these traits (courtship, dichromatism, ornamentation) present in males of each species and ranged from 0 (indicating absence of all these traits) to 3 (presence of all 3 traits).

Data on relative gonopodium length (the ratio of gonopodium length to male standard length) and an index of sexual size dimorphism in standard length (the ratio of log-transformed standard length of the larger sex to log-transformed standard length of the smaller sex subtracted from one and made negative if males are the larger sex and positive if females are the larger sex) were taken from Pollux et al.[34]. Average trait values were calculated for poeciliid species with data from multiple populations. These continuous characters were each converted to binary characters (0/1) by scoring the bottom 50% of species as small (i.e., state 0) and the top 50% as large (i.e., state 1).

**Phylogenetic tree**. For our comparative analyses, we employed the timetree of Reznick et al.[55], which contains 177 poeciliid and 116 outgroup taxa. This phylogeny was built using ML analysis of a 28-gene (20 nuclear, 8 mitochondrial) concatenated DNA matrix[34]. Molecular dating analyses were performed by integrating this molecular phylogeny with 16 primary fossil calibrations; divergence times were estimated using the mcmctree program in PAML 4.4c[56] with independent rates and hard-bounded constraints (details in Reznick et al.[55]). The drop.tip function in the package Ape[57] was used to remove species from the timetree not present in particular character analyses.

**Ancestral state reconstructions**. Ancestral states of binary characters were estimated with ML, maximum parsimony, and stochastic character mapping. The likelihoods of two transition rate models (equal rate and all rates different) were compared using the Ace function of the R package Ape[57]. Akaike Information Criterion (AIC) was used to determine the best-fitting model, which was then used to reconstruct the marginal ancestral state of the root and the scaled likelihood of

all other nodes. Parsimony ancestral state reconstructions were implemented in Mesquite version 3.51[58]. Furthermore, stochastic character mapping[59] implemented in the R package phytools[60] was used to estimate the marginal probabilities of all nodes or edge states based on joint sampling; character states were summarized from 1000 stochastic character maps.

**Correlated character evolution**. To investigate the evolution of male sexually selected traits and female provisioning of offspring, we employed Discrete evolution models implemented in a Bayesian framework in BayesTraits V3[61,62]. For each analysis, we removed all species from the tree lacking paired male and female data and scaled the tree as recommended in the manual of BayesTraits V3. All analyses employed reversible-jump MCMC with an exponential prior with mean seeded from a uniform hyperprior ranging from 0 to 20. Models are sampled in direct proportion to their fit to the data in MCMC. The reverse-jump procedure[63] allows for a reduction in model complexity and over-parametrization by setting some transition rates equal to zero or equal to one another. MCMC chains were run for 400 million iterations with a burn-in of 500,000 and sampling every 200,000 iterations. Visual inspections of traces of all parameter estimates in Tracer v1.6[64] confirmed that all chains had adequate mixing and reached convergence, with parameter estimates having effective sample sizes >1000 in all analyses. All MCMC analyses were run in triplicate; independent runs always produced qualitatively similar results.

We used Discrete independent and dependent models to evaluate the pathways by which male and female traits originate and to test the hypothesis that the evolution of female reproductive mode is driving the evolution of male traits. Discrete models require two binary traits (e.g., presence/absence of placenta; presence/absence of sexually selected trait). Under the independent model, the two traits evolve independently of one another and the model estimates four transition rates (the rate of gain and loss for each trait). Conversely, the dependent model estimates the transition rates between the combination of character states (presence/absence) that the two binary traits can jointly take and thus estimate eight transition rates[61]. If the dependent model fits the data better than the independent model, the two traits evolve in a correlated fashion. We thus compared the fit to the data of the independent and dependent models by estimating their marginal likelihood using a stepping stone sampler in MCMC in BayesTraits and calculating the Bayes factors[65]. We used a stepping stone sampler with 200 stones and 200,000 iterations per stone. Bayes factors >2 are considered positive evidence for the model with the higher harmonic mean, >5 as strong evidence, while values >10 as very strong evidence[61]. When supported as the best-fitting model, the dependent model can reveal the supported evolutionary pathways from the absence to the presence of both traits and whether a given combination of character states for the two variables is evolutionary stable. For characters exhibiting dependent evolution, we calculated the percentage of models in the posterior distribution in which each transition rate had a value of zero. These Z-scores[61] allow for examination of the contingency of evolutionary transitions among character pairs; transition rates with low Z-scores indicate likely evolutionary pathways. Thus it can be evaluated whether state changes in one character are contingent on the state of another character.

Two additional checks were performed on the robustness of the BayesTraits results. First, we tested the effect that fixing the root state had on model output. Second, we performed a sensitivity analysis to determine how the choice of criterion used to define long versus short gonopodia and high versus low sexual dimorphism index standard length influenced these models. Further details can be found in Supplementary Note 3 (Supplementary Tables 6–8, Supplementary Figs. 16–18).

**Diversification analyses**. We performed character-dependent diversification analyses for five traits of interest in the subfamily Poeciliinae. Placentotrophy, courtship, sexual dichromatism, and ornamentation were scored as binary characters (presence/absence) and were each analyzed using a binary state speciation and extinction (BiSSE) model. The sexual selection index is a multi-state character and was analyzed with a multiple state speciation and extinction (MuSSE) model. Analyses were conducted in the R package Diversitree[39]. Outgroup taxa were pruned at the base of the subfamily Poeciliinae. Six duplicate populations of the same poeciliid species were removed, creating a species-level phylogeny. To account for incomplete taxon sampling, we used two alternative procedures: sampling fraction and unresolved tips. In the sampling fraction method, we specified the number of species with character data in the phylogeny out of the total described species in the subfamily Poeciliinae. This procedure assumes that the tree contains a random sample of extant species with respect to character state; this assumption is likely valid given the wide taxonomic coverage (nearly all genera represented) and because species were not chosen for inclusion based on character state. In the unresolved tips method, species not included in the phylogeny were placed into an existing unresolved tip clade (at or below generic level) on the basis of taxonomy. Both analyses yielded qualitatively similar results (Supplementary Table 9), so only those of the sampling fraction method are presented in the main text. We estimated state-specific speciation and extinction rates using BiSSE or MuSSE models in a Bayesian framework. Priors for each parameter followed an exponential distribution with rate $1/(2r)$ where $r$ is the character-independent diversification rate, and ML-estimated model parameters served as a starting point.

MCMC chains were run for 10,000 generations with the first 10% discarded as burn-in. We also compared the fit of alternative models in an ML framework using AIC and lnLik (by means of Chi-square test).

Concerns have been raised regarding a high type 1 error rate (false positives) in character-dependent diversification analyses[40]. We addressed this potential source of error by first estimating the predicted level of type 1 error in our data set using simulations. Given the real tree and a hypothetical trait with two states and transition probabilities between the two states derived from the real data, we simulated the evolution of each trait 100 times, each time beginning with the predicted ancestral state, which is the absence of the trait. We compared two models, one in which the speciation and extinction rates of each character state are forced to be equal and one in which they are allowed to be different from one another. For each simulated outcome, we evaluated the data for the fit of the two models, assigned the probability that the unequal rates model was a better fit, and then generated a frequency distribution for these probabilities. We expect that there should be a significant difference only 5% of the time. We actually observed apparent significance in far >5% of the simulated trees, so there was clear evidence of type 1 errors when applying diversitree. We then used this frequency distribution of $p$ values to determine what $p$ value corresponds to a 5% cut off. If the observed $p$ value from the empirical data is within the 5th percentile of significance (i.e., more extreme than 95 out of 100 $p$ values generated from the model comparison on the simulated output), then this is taken as evidence of significant character-dependent diversification. For each binary trait, we also performed a nonparametric FiSSE (Fast, intuitive State-dependent Speciation-Extinction) analysis utilizing the script in Rabosky and Goldberg[41].

Lastly, we implemented joint analyses on the effect of male sexually selected traits and placentotrophy on diversification using the diversitree "make.musse. multitrait" command. For each pair of traits, we first fit a full model including "main effects" of the two traits on speciation and extinction and an independent model of character evolution (i.e., four transition rate parameters). We then fit two constrained models in which the main effect of each trait on speciation was deleted (i.e., fixed at 0). We compared the fit of these full and constrained models in an ML framework using AIC and lnLik (by means of Chi-square test). These model comparisons were designed to test whether male sexually selected traits or female reproductive mode is a better predictor of speciation rate. In addition to ML, the full model was fit in a Bayesian MCMC framework. Priors for each parameter followed an exponential distribution with rate $1/(2r)$ where $r$ is the character-independent diversification rate[39]. MCMC chains were run for 10,000 generations with the first 10% discarded as burn-in.

**Analyses of the MI as a continuous trait**. We performed ancestral state reconstructions, correlated evolution analyses, and diversification analyses with the placenta scored as a binary presence or absence character. However, it is also possible to perform similar analyses using the continuous MI as a quantitative measure of the degree of placentation or lack thereof. In the Supplemental Materials, we report the results of these analyses and discuss the merits and limitations of both approaches (Supplementary Tables 14 and 15, Supplementary Figs. 20–22). Briefly, ancestral state reconstructions allow us to make interpretations regarding the root state of the subfamily and how many times placentation evolved; here we think it may be more biologically realistic to use the presence/absence of placentation rather than the continuous MI (Supplementary Fig. 20), because, among other reasons, doing so allows for the inclusion of non-livebearing outgroup taxa, which is important for accurate reconstructions. Next, we implemented correlated evolution analyses using phylogenetic logistic regression[66,67] to evaluate the relationship between the continuous MI and binary male traits[34]. Comparison with BayesTraits Discrete analyses, in which placentation is treated as a binary character, reveal that the results of these two modeling efforts concord well (Supplementary Fig. 21, Supplementary Table 14). However, the use of BayesTraits Discrete models (which require two binary traits) allow us to go a step beyond testing for correlation and make inferences regarding the order of evolution of male sexually selected traits and the placenta. Lastly, to evaluate the relationship between the MI and diversification rates, we implemented QuaSSE (quantitative state speciation and extinction) models[68] in the R package diversitree[39]. Overall, the best-fitting QuaSSE model supports interpretations derived from our BiSSE analysis in which we tested for a correlation between speciation rate and the presence or absence of the placenta (Supplementary Table 15, Supplementary Fig. 22).

**Reporting summary**. Further information on research design is available in the Nature Research Reporting Summary linked to this article.

## Data availability
The dataset generated and analyzed during the current study is available in the Dryad Digital Repository: https://doi.org/10.5061/dryad.7g5b162.

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

## Acknowledgements

We thank Sally Otto for advice on the implementation of Diversitree, Andrew Meade and Isabella Capellini for advice on the implementation of BayesTraits, and Locke Rowe for commenting on the manuscript and advice on the interpretation of some of the results. Yuridia Reynoso played a key role in performing the dissections associated with the characterization of maternal provisioning. This work was supported by the Academic Senate of the University of California, Riverside, the University of Hull, and by the National Science Foundation (University of California—DEB-0416085, DEB-1754669, and PRFB Award 1523666; Montclair State University—DEB-1556701, DBI-1725932). B.J.A.P. was supported by VIDI grant 864.14.008 of the Netherlands Organisation for Scientific Research (NWO).

## Author contributions

D.N.R. designed the study. B.J.A.P. and D.N.R. collected the data. R.W.M. and M.S.S. constructed the phylogenetic trees. A.I.F. organized the data and conducted analyses. D.N.R. and A.I.F. wrote the manuscript with input from all authors.

## Additional information

**Competing interests:** The authors declare no competing interests.

