## [Peer Review File · Nature Communications]

Reviewers' Comments:

Reviewer #1:

Remarks to the Author:

Overall I find this to be an outstanding paper that provides novel insights into the coevolutionary relationships of female reproductive mode, male sexually selected traits, and rates of speciation. The analyses are solid and the conclusions well-drawn, and the main results represent a substantive contribution to an important literature.

A number of points need to be addressed, however, to improve the article:

- (1) The logic behind how male sexually selected traits are expected to be related to female provisioning could be made more transparent, in the Abstract and Introduction, especially in the context of validated costs and benefits to females, and to males.
- (2) line 58, IS associated
- (3) The word 'governing' is used many times in the article (I think too often), and its meaning should be made more explicit.
- (4) The first paragraph of the Introduction is much too long.
- (5) line 84. "adaptation and conflict can act as accomplices in causing the evolution of " I do not think that Crespi and Semeniuk said quite this, even if it is not quite clear what the statement means. Please use more precise language here.
- (6) line 86. Please explain in more detail the 'paternal influence, ' here , and describe whether and how such influence have in fact been demonstrated. This point is salient to the predictions addressed.
- (7) lines 99-100. Has this prediction ever been made before?
- (8) lines 160-1. Please describe the importance of superfetation to the predictions and tests.
- (9) line 216 "An unexpected feature of these results is that male sexually selected traits are almost as likely to be lost as gained within non-placental lineages ". Why is this finding unexpected? What is its meaning in the context of the other findings? Why should it be lost in these conditions?
- (10) Line 267 and later. "In two of three analyses, models that included both male and female main effects or only the male main effect were statistically indistinguishable in their ability to predict speciation rate. Models that included only maternal provisioning were significantly and substantially worse than the other two models in their ability to predict speciation rate. These results strongly support the argument that male traits alone govern the " I do not see how support can be so strong of the finding is made in only 2 of 3 cases? Please clarify.
- (11) "rate of speciation and the rate of speciation accelerates as males acquire traits associated with sexual selection (Fig. 7). These results in turn suggest that the evolution of pre-copulatory reproductive isolation plays a much more important role than post-copulatory reproductive isolation in causing the origin of new species in Poeciliinae. " . How did the authors evaluate or quantify post-reproductive isolation, then, given that they are comparing per- and post-? Is there evidence that placentation affects the evolution of post-zygotic isolation in these fishes?

(12) The Conclusions section is very long and includes material other than conclusions.

(13) Typo that jumped out at me; likely to be others: Elliot, M.G. and B.J. Crespi, Placental invasiveness **edmiates the evolution of hybrid inviability in mammals. American Naturalist, 2006.

(14) Figure 3, the inset 'Family-Wide Ancestral State'. It is not clear how 'probably' can apply well to the 4/5 pie charts that show near 50:50 proportions.

These points are relatively minor and do not substantially take away from the very high quality of this work overall.

Bernie Crespi

Reviewer #2:

Remarks to the Author:

In manuscript NCOMMS-19-05606 („How Conflict Shapes Evolution in Poeciliid Fishes”), Furness and colleagues examine the transitions male sexually selected traits and female placentation in poeciliid fishes. They test an extended version of the viviparity-driven conflict hypothesis (VDCH), predicting that females producing large eggs (i.e., investment pre-fertilization) should be choosier with regard to mates, and so give rise to more elaborate male sexual traits than in females that invest in their offspring primarily after fertilization (in the form of placentation).

The authors find that, indeed, at least some of the male sexually selected traits show divergent evolution primarily in those species that lack female placentation, with the state of placentation preceding diversification in male traits.

If found this study very thoroughly executed, using state-of-the-art phylogenetic tools, the results are intriguing, and the paper seems transparent and well written, including clear and appealing illustrations. I believe this paper will find a broad readership.

I did not find any severe issues but do have a few suggestions for clarification.

1. The authors state that sexual size dimorphism is a male signaling trait and that females are larger. In other words, sexual selection on males should result in a decrease in SSD in taxa with female-biased SSD (i.e., male size approaching that of females), whereas fecundity selection might drive the evolution of larger female size. The authors seem to treat and discuss their SDISL similarly as sexual dichromatism or courtship, meaning that bigger is better (which is the typical interpretation if SSD is male-biased). As I understand, however, the absence of placentation gives rise to increased SSD, which, if true, suggests to me that this may not be through sexual selection on the males but rather enhanced selection on females, possibly linked to producing larger eggs. And once females are relatively large, placentation evolves (red arrow in Fig. 4). Please clarify.

2. Since poeciliid fishes are extremely colorful and ornamented, including females, the estimate of sexual dichromatism as a sexually selected trait may not be reliable. There is mounting evidence that sexual (e.g. male choice) and other forms of selection can drive the evolution of bright female coloration and ornamentation (mostly seen in birds, but also in fishes). In other words, a species assigned here as sexually monochromatic does not mean that sexual selection is weaker than in a

dichromatic species (in monochromatic species, both sexes could simply be brighter than in a dichromatic one). Clearly, the overall results may not change significantly, but the authors should critically evaluate their assignment to mono- and dichromatism and justify the reliability of this binary assignment.

3. Are there any species in which both sexes are ornamented per the authors' definition of ornamentation? If so, how were these treated with regards to the evolution of male traits? Is it possible that discrepancies in the assignment of ornamentation might have caused the "negative" results for this trait?

4. Was the binary scoring of gonopodium length based on absolute or relative gonopodium length? In my view, it should be the latter.

5. Throughout paper (and figures), I recommend using the term "sexual dichromatism" as there are also other forms of dichromatism

And here two minor edits:

1. On line 67, please delete "a" before sea urchins

2. In Fig. 3, note that the absence/presence bars are not aligned for dichromatism

Reviewer #3:

Remarks to the Author:

This manuscript presents a phylogenetic comparative analysis that tests the links between reproductive mode, sexual selection, and speciation rate. Taking advantage of the repeated origins of the placenta in poeciliid fishes and applying insights about microevolutionary processes that govern the relationship between maternal investment, sexual selection, and reproductive isolation, the authors test several predictions about macroevolutionary patterns. The major result that mode of maternal investment influences the rate of evolution of male sexually selected traits is compelling, and the rarity of male traits in lineages with placentation is striking. The manuscript is well written and the analyses are quite thorough.

My major concern with this work is the conversion of Matrotrophy Index, a continuously varying trait, into a discrete trait and then modeling its evolution as a Markov process. My concern here is that according to the Markov model, the transition rate between lecithotrophy and placentotrophy will be equal across lineages regardless of whether they are characterized by MIs that are close to or far from the threshold separating them. This does not seem like a desirable property and the binary character could miss some important variation among lineages.

One consideration that may clear up the issue is whether crossing the threshold corresponds to an evolutionary origin of "integration of specialized embryonic and maternal tissues in the placenta," which is the biological definition of placentotrophy offered by the authors (lines 80-81). Does the chosen MI threshold generally occur when mother provisions for her offspring through a placenta? This matters, I think, because the choice to model maternal investment evolution as a discrete variable rather than a continuous one is justified if the threshold reflects a real biological difference in how mothers provision for their young. But if the threshold is based simply on the distribution of MI values among species, then it seems more appropriate to model MI as a continuous trait. In this case, the authors could use phylogenetic logistic regression (Ives and Garland 2010; Syst Biol) to evaluate the

effects of MI on male traits and QuaSSE (FitzJohn 2010, Syst Biol) to evaluate the relationship between MI and diversification rates.

A few additional points of clarification.

1) I was not entirely clear on the basis for the expectation that females would be less choosy in placental species. Even though the ova receive less provisioning upfront, wouldn't there still be a large cost associated with allowing fertilization by an inferior male? I think the authors are saying that females are less choosy at copulation but then select which sperm fertilizes the egg, but I was not completely sure on this point. Also, this post-copulatory selection seems to be more tied to internal fertilization than placentation. Do only placental species use post-copulatory mate selection?

2) In addition to maternal investment's effect on male traits, the authors also evaluate the prediction that male traits affect provisioning, but the biological basis for this scenario is not clear. I recognize that the results offer no support for this pattern, but in some ways it seems like the authors have set up these patterns as alternative outcomes of competing hypotheses, though the alternative hypothesis (sexual selection spurs transitions in maternal investment) is not clearly stated.

3) What proportion of species diversity for poeciliinae has been sampled for this study, and how much overlap in sampling is there across variables? The authors account for incomplete species sampling in their analyses of speciation rates, but I would like the authors to comment on the robustness of these results given the proportion of species sampled for this analysis.

We thank all three reviewers for their constructive comments. Below we address each of them, with our responses indicated in red font.

Response to Reviews:

Reviewers' comments:

Reviewer #1 (Remarks to the Author):

Overall I find this to be an outstanding paper that provides novel insights into the coevolutionary relationships of female reproductive mode, male sexually selected traits, and rates of speciation. The analyses are solid and the conclusions well-drawn, and the main results represent a substantive contribution to an important literature.

A number of points need to be addressed, however, to improve the article:

(1) The logic behind how male sexually selected traits are expected to be related to female provisioning could be made more transparent, in the Abstract and Introduction, especially in the context of validated costs and benefits to females, and to males.

We have rewritten the introduction by splitting the long opening paragraph into three. We expanded the reference to Crespi and Semeniuk [1] into a full paragraph with added detail on the costs and benefits to females and offspring (as bearers of the male haploid genome). The following paragraph deals more explicitly with the role of males. The traditional way of describing intergenomic conflict is in the relatedness of offspring to mothers and to each other, but a different way of stating the conflict is to say that the offspring are a vehicle for the expression of the paternal genome. The quantity of publications that support the “validated costs and benefits to females, and to males” is large. We have used the Crespi and Semeniuk [1] paper as a surrogate for that literature. We also worked with other versions of the abstract but in the end felt that the original suited our needs, given the length limitations.

(2) line 58, IS associated - **Done**

(3) The word 'governing' is used many times in the article (I think too often), and its meaning should be made more explicit.

We eliminated the use of this word in the context of diversification analyses. We now only use the word when describing the relationship between female reproductive mode and male sexually selected traits. We do so to highlight the inferred causality of the relationship (i.e. to convey more than a correlation between the traits). Specifically, our results suggest it is female reproductive mode (i.e. placentation) that is the determinant of how male traits evolve.

(4) The first paragraph of the Introduction is much too long.

We subdivided it into three paragraphs. The reviewer is correct, not just about length, but about

the original paragraph having contained a composite of three themes that should have been presented separately. As noted in point 1) above, these changes improve the transparency and readability of our introduction.

5) line 84. "adaptation and conflict can act as accomplices in causing the evolution of " I do not think that Crespi and Semeniuk said quite this, even if it is not quite clear what the statement means. Please use more precise language here.

We agree. The reference to Crespi and Semeniuk [1] became the third paragraph. There, we expanded on the rather cryptic reference to Crespi and Semeniuk [1] in the original version of the paper and hope that our treatment accurately reflects the message of their paper.

(6) line 86. Please explain in more detail the 'paternal influence,' here, and describe whether and how such influence have in fact been demonstrated. This point is salient to the predictions addressed.

We modified this line and made more extensive reference to Crespi and Semeniuk. At this point in time, we have used that paper as a surrogate for “whether and how such influence has in fact been demonstrated.”

(7) lines 99-100. Has this prediction ever been made before?

It is our extrapolation of ideas presented by Zeh and Zeh [2]. These authors suggest that “rather than relying on conventional phenotype-based mate choice, [viviparous] females may reduce the risk and/or cost of fertilization by genetically-incompatible sperm more effectively by mating with more than one male and exploiting the postcopulatory mechanisms of sperm competition, female choice of sperm and the reallocation of maternal resources from defective to viable embryos. Such selection for polyandry and decreased reliance on precopulatory female choice...” [p. 940]. This evolutionary shift from oviparity to viviparity [which we interpret as the shift from lecithotrophy to matrotrophy] should correlate with (1) increased levels of polyandry, (2) decreased reliance on precopulatory sexual selection, and (3) increased reliance on postcopulatory sexual selection.

(8) lines 160-1. Please describe the importance of superfetation to the predictions and tests.

Zeh and Zeh predicted that viviparity (placentation) will favor polyandry. Superfetation is known to promote polyandry in some mammals. In mammals, it is manifested by ovulation and mating in the fall, then delayed implantation. They ovulate a second time in the spring, mate a second time, then both sets of fertilized eggs implant together. On this basis, we predict that the evolution of superfetation will be associated with the evolution of placentation in the Poeciliid fishes if it functions as in mammals to promote polyandry. This is indeed the case [3]. We added a brief statement to this effect in the new version of the manuscript. In truth, we do not know if superfetation in fish has the same effect as it does in mammals. But, we do know how to do an experiment to test for this possibility and hope to execute such an experiment soon.

Our extension of the viviparity-driven conflict hypothesis predicts a relationship between mode of female provisioning and male sexually selected traits. The hypothesis doesn't make any

predictions regarding superfetation and male sexually selected traits which is why we didn't test for them.

(9) line 216 "An unexpected feature of these results is that male sexually selected traits are almost as likely to be lost as gained within non-placental lineages". Why is this finding unexpected? What is its meaning in the context of the other findings? Why should it be lost in these conditions?

Good question. The prior literature on this topic does not offer a firm answer. Weins' discussion is an elaborate way of saying that sexual and natural selection are often in conflict, such that a trait that is favored by sexual selection can be disfavored by natural selection. This is not a radical proposition. If and when this is true, it sets up the same pattern of antagonistic evolution as seen in the evolution of the placenta, as argued in Crespi and Semeniuk [1], and creates the same expectation of a dynamic ebbing and waning of sexually-selected traits in males. We did not make any changes in this part of the manuscript and instead modified the discussion (lines 310-315) to better articulate why we think the evolution of such traits are readily reversed.

(10) Line 267 and later. "In two of three analyses, models that included both male and female main effects or only the male main effect were statistically indistinguishable in their ability to predict speciation rate. Models that included only maternal provisioning were significantly and substantially worse than the other two models in their ability to predict speciation rate. These results strongly support the argument that male traits alone govern the"

I do not see how support can be so strong of the finding is made in only 2 of 3 cases? Please clarify.

The make.musse.multitrait analyses (i.e. Figure 7 and Table S13) show that each male sexually selected trait is associated with higher rate of speciation and placentation is not, when both traits are included in the model. This can be seen most readily in Figure 7 which presents the Bayesian mcmc posterior distributions of speciation 'main effects' estimated from these models. In all 3 analyses presented in Figure 7 the 95% credibility interval of λ_P is overlapping 0, and that of $\lambda_{C/D/O}$ is significantly higher than 0 with none of the interval overlapping 0. The interpretation of these results is that placentation doesn't significantly affect speciation rate, while each male trait (courtship, dichromatism, and ornamentation) is associated with an increase in speciation rate. We clarified this in the text (lines 284-294), and moved the more detailed explanation of the complementary maximum likelihood model comparison results to the supplementary materials, which is where such results are presented (see below).

In addition to fitting the make.musse.multitrait models using Bayesian mcmc (Figure 7), these analyses were also performed in a maximum likelihood model comparison framework (Table S13). For each pair of traits, we first fit the full model including 'main effects' of the two traits on speciation and extinction and an independent model of character evolution (i.e. 4 transition rate parameters). We then fit two constrained models in which the main effect of each trait on speciation was deleted (i.e. fixed at 0). These model comparisons were designed to test whether male sexually selected traits or female reproductive mode is a better predictor of speciation rate. The maximum likelihood model comparisons for all 3 pairs of male and female traits support the interpretation that male traits are associated with significantly higher speciation rate. In 2 of 3 of

these analyses, placenta was associated with no difference in speciation rate and in the third (placenta-ornamentation) placenta was associated with lower speciation rate.

Below are the Maximum likelihood parameter estimates from the make.musse.multitrait analyses, with those of interest highlighted in yellow.

Courtship-Placenta

lambda0	lambdaC	lambdaP	mu0	muC	muP	qC01.0	qC10.0	qP01.0	qP10.0
2.415	25.97	1.002	4.345	10.283	-4.345	0.876	9.118	0.534	0

Dichromatism-Placenta

lambda0	lambdaD	lambdaP	mu0	muD	muP	qD01.0	qD10.0	qP01.0	qP10.0
7.191	14.904	-3.625	10.128	0	-10.128	0.539	6.61	0.421	0

Ornamentation-Placenta

lambda0	lambdaO	lambdaP	mu0	muO	muP	qO01.0	qO10.0	qP01.0	qP10.0
11.197	30.558	-6.236	15.265	-0.92	-14.344	0	25.327	0.293	0

In the analysis of Courtship-Placenta and Dichromatism-Placenta, the ML estimate of lambdaC and lambdaD (the main effects of courtship and dichromatism on speciation rate) were estimated as 25.970 and 14.904, respectively; while lambdaP (the main effects of placenta on speciation rate) were estimated as 1.002 and -3.625, respectively. In the two constrained models each of these parameters were fixed at 0 (equivalent to deleting the parameter) and the log likelihood of each model was compared to the full model. When lambdaC (25.970) or lambdaD (14.904) were fixed at 0 the model exhibited substantially poorer fit – indicating that species with courtship and dichromatism exhibit significantly higher speciation rate. In contrast, fixing lambdaP (1.002 and -3.625 respectively) at 0 resulted in no difference in model fit compared to the full model – indicating that species with placenta exhibited neutral / non-significant speciation rate. In the analysis of Ornamentation-Placenta, lambdaO (main effect of ornamentation on speciation rate) was equal to 30.558, while lambdaP (main effect of placenta on speciation rate) was equal to -6.236. Unsurprisingly, the model in which lambdaO was fixed at 0 exhibited poor fit because species with ornamentation have high speciation rate (30.558). However, because lambdaP was slightly negative (-6.236), fixing this parameter at 0 resulted in a significantly poorer fit than when the parameter was freely estimated from the data – indicating that species with placenta exhibited significantly lower speciation rate. Therefore, the interpretation of all three maximum likelihood model comparisons is similar – the male trait is associated with (significantly) higher speciation rate and placenta with neutral or (significantly) lower speciation rate. This more detailed explanation of these results was added to the supplementary materials (in the caption to Table S13). These maximum likelihood model comparisons support the interpretation derived from the Bayesian mcmc analyses – that it is the male sexually selected traits associated with increased speciation rates and the placenta is not.

(11) "rate of speciation and the rate of speciation accelerates as males acquire traits associated

with sexual selection (Fig. 7). These results in turn suggest that the evolution of pre-copulatory reproductive isolation plays a much more important role than post-copulatory reproductive isolation in causing the origin of new species in Poeciliinae."

How did the authors evaluate or quantify post-reproductive isolation, then, given that they are comparing per- and post-? Is there evidence that placentation affects the evolution of post-zygotic isolation in these fishes?

There is only limited evidence at this time. The best work is the series of papers by Schrader and Travis on *Heterandria formosa* in which they demonstrate substantial post-zygotic RI among closely related populations. In their case it is driven by differences among populations in offspring size. Otherwise, ours is the same hypothesis articulated by Zeh and Zeh [2] – If placentation causes an increase in the rate of evolution of post-copulatory reproductive isolation then it should also cause accelerated speciation.

(12) The Conclusions section is very long and includes material other than conclusions.

Okay. One way to deal with this issue is to relabel this section as "Discussion and Conclusions", which we have done. Otherwise, we think this section is okay as a way of ending the paper.

(13) Typo that jumped out at me; likely to be others: Elliot, M.G. and B.J. Crespi, Placental invasiveness **edmiates the evolution of hybrid inviability in mammals. American Naturalist, 2006.

We made this correction plus edited the rest of the literature cited. Thank you for catching the error.

(14) Figure 3, the inset 'Family-Wide Ancestral State'. It is not clear how 'probably' can apply well to the 4/5 pie charts that show near 50:50 proportions.

Here we use the term probably to refer to 'the most likely' ancestral state. Admittedly, 4 of 5 male traits exhibit a lot of uncertainty in the maximum likelihood ancestral state reconstruction of the root of the family, which we state in the main text. However, we are not really sure what word to replace 'probably' with here. Note that the 'most likely states' are fully congruent with the parsimony ancestral state reconstructions (Table S4) and that we also summarize the results of the parsimony reconstructions in the text.

These points are relatively minor and do not substantially take away from the very high quality of this work overall.

Bernie Crespi

Reviewer #2 (Remarks to the Author):

In manuscript NCOMMS-19-05606 („How Conflict Shapes Evolution in Poeciliid Fishes”), Furness and colleagues examine the transitions male sexually selected traits and female

placentation in poecilid fishes. They test an extended version of the viviparity-driven conflict hypothesis (VDCH), predicting that females producing large eggs (i.e., investment pre-fertilization) should be choosier with regard to mates, and so give rise to more elaborate male sexual traits than in females that invest in their offspring primarily after fertilization (in the form of placentation).

The authors find that, indeed, at least some of the male sexually selected traits show divergent evolution primarily in those species that lack female placentation, with the state of placentation preceding diversification in male traits.

I found this study very thoroughly executed, using state-of-the-art phylogenetic tools, the results are intriguing, and the paper seems transparent and well written, including clear and appealing illustrations. I believe this paper will find a broad readership.

I did not find any severe issues but do have a few suggestions for clarification.

1. The authors state that sexual size dimorphism is a male signaling trait and that females are larger. In other words, sexual selection on males should result in a decrease in SSD in taxa with female-biased SSD (i.e., male size approaching that of females), whereas fecundity selection might drive the evolution of larger female size. The authors seem to treat and discuss their SDISL similarly as sexual dichromatism or courtship, meaning that bigger is better (which is the typical interpretation if SSD is male-biased). As I understand, however, the absence of placentation gives rise to increased SSD, which, if true, suggests to me that this may not be through sexual selection on the males but rather enhanced selection on females, possibly linked to producing larger eggs. And once females are relatively large, placentation evolves (red arrow in Fig. 4). Please clarify.

Pollux et al. [3] show that the significant positive relationship between the matrotrophy index and the magnitude of sexual size dimorphism is caused by a decrease in male body size rather than an increase in female body size. The ancestral state, and the predominant state in most species, is for females to be larger than males. We think this is likely a consequence of fecundity selection, as suggested by the reviewer. But, some species of Poeciliinae with highly ornamented males also have males that are larger than females or have reduced size differences between males and females. What we are proposing here then is that an increase in male size at maturity in these non-placental lineages will result first in a decrease in the size difference between males and females and, in some cases, may actually cause a reversal of the dimorphism, meaning that males can be larger than females. We made some refinements to this part of the text with the hope of making it clearer.

2. Since poecilid fishes are extremely colorful and ornamented, including females, the estimate of sexual dichromatism as a sexually selected trait may not be reliable. There is mounting evidence that sexual (e.g. male choice) and other forms of selection can drive the evolution of bright female coloration and ornamentation (mostly seen in birds, but also in fishes). In other words, a species assigned here as sexually monochromatic does not mean that sexual selection is weaker than in a dichromatic species (in monochromatic species, both sexes could simply be brighter than in a dichromatic one). Clearly, the overall results may not change significantly, but

the authors should critically evaluate their assignment to mono- and dichromatism and justify the reliability of this binary assignment.

Your point is well taken. Poeciliid species scored as exhibiting dichromatism all exhibit the same pattern - brightly colored males and less brightly colored females. In poeciliid species scored as having no dichromatism, both sexes exhibit the same or similar monochromatic coloration and are both drab (i.e. not both brightly colored). We aren't aware of any poeciliid species where females are more colorful than males or where males and females are both colorful. This statement is based on D. Reznick's 40 years of personal observations on members of the family, which includes having worked with live specimens of nearly 100 species plus museum collections of over 50 additional species.

3. Are there any species in which both sexes are ornamented per the authors' definition of ornamentation? If so, how were these treated with regards to the evolution of male traits? Is it possible that discrepancies in the assignment of ornamentation might have caused the "negative" results for this trait?

No, there are no species in which both sexes are ornamented. Ornamentation (sword-like extension of the tail, sail-like dorsal fin, moustache-like filamentous extension of the upper lip, and extreme lateral body compression) were only found in males.

The lack of correlation between presence / absence of placentation and presence / absence of ornamentation is almost certainly due to the limited sample size of species with ornamentation. However, even though non-significant, the pattern seen between these two variables is nearly identical for the other traits. Specifically, all origins of ornamentation are in non-placental lineages (*Xiphophorus*, and *Poecilia*).

4. Was the binary scoring of gonopodium length based on absolute or relative gonopodium length? In my view, it should be the latter.

The measure of gonopodium length was relative. Gonopodium length was divided by standard length to give a relative measure, facilitating comparison among species of different body size. This relative measure of gonopodium length was then converted to binary scoring (short / long). This is now clarified in the text.

5. Throughout paper (and figures), I recommend using the term "sexual dichromatism" as there are also other forms of dichromatism

We now clearly specify sexual dichromatism in the text and in the figure legends. However, in the figures themselves, where space is limited, we feel that it is better to keep things simple and use the term 'Dichromatism' instead of 'Sexual Dichromatism'. Furthermore, in the figures the other sexually selected male traits are described by a single word – i.e. Courtship, Ornamentation.

And here two minor edits:

1. On line 67, please delete “a” before sea urchins

Done.

2. In Fig. 3, note that the absence/presence bars are not aligned for dichromatism

This was fixed. Thank you.

Reviewer #3 (Remarks to the Author):

This manuscript presents a phylogenetic comparative analysis that tests the links between reproductive mode, sexual selection, and speciation rate. Taking advantage of the repeated origins of the placenta in poeciliid fishes and applying insights about microevolutionary processes that govern the relationship between maternal investment, sexual selection, and reproductive isolation, the authors test several predictions about macroevolutionary patterns. The major result that mode of maternal investment influences the rate of evolution of male sexually selected traits is compelling, and the rarity of male traits in lineages with placentation is striking. The manuscript is well written and the analyses are quite thorough.

My major concern with this work is the conversion of Matrotrophy Index, a continuously varying trait, into a discrete trait and then modeling its evolution as a Markov process. My concern here is that according to the Markov model, the transition rate between lecithotrophy and placentotrophy will be equal across lineages regardless of whether they are characterized by MIs that are close to or far from the threshold separating them. This does not seem like a desirable property and the binary character could miss some important variation among lineages.

One consideration that may clear up the issue is whether crossing the threshold corresponds to an evolutionary origin of “integration of specialized embryonic and maternal tissues in the placenta,” which is the biological definition of placentotrophy offered by the authors (lines 80-81). Does the chosen MI threshold generally occur when mother provisions for her offspring through a placenta? This matters, I think, because the choice to model maternal investment evolution as a discrete variable rather than a continuous one is justified if the threshold reflects a real biological difference in how mothers provision for their young. But if the threshold is based simply on the distribution of MI values among species, then it seems more appropriate to model MI as a continuous trait. In this case, the authors could use phylogenetic logistic regression (Ives and Garland 2010; Syst Biol) to evaluate the effects of MI on male traits and QuaSSE (FitzJohn 2010, Syst Biol) to evaluate the relationship between MI and diversification rates.

There is a real biological difference between lecithotrophic and placentotrophic species. However, as a practical matter, defining where the threshold lies is challenging. Egg-laying species are reported to have eggs which lose ~30% of their mass during development because of costs of metabolism. So presumably a MI value higher than 0.7 in a live-bearing species is indicative of some degree of matrotrophy. The matrotrophy index is a population or species level

measure generated through the aggregation of data from individual pregnant females. Therefore, its measurement is dependent upon adequate sample sizes and a good distribution of embryo developmental stages, as well as the variance in the data. It is for this reason that we chose a statistical approach to defining the threshold between lecithotrophy and placentotrophy that takes into account these factors. Furthermore, we took a conservative approach such that species with an MI at the border between lecithotrophy and matrotrophy were scored as non-placental.

Your point about performing analyses on the continuous Matrotrophy Index is well taken. In this manuscript we performed ancestral state reconstructions, correlated evolution analyses, and diversification analyses with the placenta scored as a binary presence or absence character. However, it is also possible to perform similar analyses using the continuous Matrotrophy Index as a measure of the degree of placentation or lack thereof. We have now completed these analyses. For the most part they are congruent with the current analyses in which we used placenta presence / absence. For ancestral state reconstructions we think use of the placenta as a binary trait is preferable, and for the BayesTraits analyses binary traits are required. For the diversification analyses we think the placenta as a binary trait versus using the continuous MI are equally justifiable (and give similar results).

We added these analyses to the supplemental materials and add a brief note discussing this point to the main text. We chose this approach so as to keep the main text consistent by presenting all analyses on placenta presence or absence. However, a specialist reader who is familiar with poeciliid fishes may also have the same questions regarding the use of the continuous Matrotrophy Index. We feel the manuscript is strengthened by inclusion of these analyses, along with discussion of the merits and limitations of both approaches, and our justifications for doing the analyses with the placenta as a binary trait. These analyses can be found in the Supplementary Materials (pp. 31-36, Tables S14-15, Figures S20-22). We hope these address all of the reviewer's concerns.

A few additional points of clarification.

1) I was not entirely clear on the basis for the expectation that females would be less choosy in placental species. Even though the ova receive less provisioning upfront, wouldn't there still be a large cost associated with allowing fertilization by an inferior male? I think the authors are saying that females are less choosy at copulation but then select which sperm fertilizes the egg, but I was not completely sure on this point. Also, this post-copulatory selection seems to be more tied to internal fertilization than placentation. Do only placental species use post-copulatory mate selection?

It's not necessarily that placental species will be less choosy, but that placental species are predicted to exhibit a shift from being choosy pre-copulation to being choosy post-copulation. Post-copulatory sexual selection can entail both cryptic female choice of which sperm fertilizes the egg (i.e. after copulation but before fertilization) and also choice after fertilization (referred to as post-zygotic sexual selection). This could take the form of selective embryo abortion or the reallocation of maternal resources from defective to viable embryos. Post-zygotic sexual

selection is only available to placental species.

Lecithotrophic species produce large energetically costly eggs, that are fully yolked prior to fertilization. Lecithotrophic females are predicted to gain the greatest reproductive benefit by being choosy, i.e. by carefully selecting their potential mate before copulation, e.g. based on male phenotype or courtship behavior. In contrast, placental species produce tiny energetically cheap eggs. They might obtain most reproductive benefits by using a different strategy, namely by (over)producing many eggs and then mating (perhaps randomly meaning that they are being less choosy, but possibly with some degree of pre-copulatory selection) with multiple males and relying on post-copulatory or even post-zygotic mechanisms of sexual selection.

Zeh & Zeh [2, 4] argue that the evolution of viviparity [but in reality, they mean matrotrophy] should correlate with a decreased reliance on pre-copulatory sexual selection and an increased reliance on post-copulatory sexual selection. Of course, in any species that has internal fertilization there is an opportunity for sexual selection after copulation but before fertilization (e.g. through sperm competition), but Zeh & Zeh [2, 4] predict that this will be more intense in matrotrophic lineages. And, importantly, post-zygotic selection (i.e. the differential allocation of maternal resources to embryos based on genotype) is only possible in placental species. So, even if post-copulatory but pre-fertilization selection is similar for lecithotrophs and matrotrophs, matrotrophic species still have opportunities for post-zygotic sexual selection that lecithotrophs do not have.

We now clarified these points in the introduction.

2) In addition to maternal investment's effect on male traits, the authors also evaluate the prediction that male traits affect provisioning, but the biological basis for this scenario is not clear. I recognize that the results offer no support for this pattern, but in some ways it seems like the authors have set up these patterns as alternative outcomes of competing hypotheses, though the alternative hypothesis (sexual selection spurs transitions in maternal investment) is not clearly stated.

This hypothesis was suggested by Haig [5] in a commentary on Pollux et al. [3]. The idea is that placentation can be brought about by males fertilizing female eggs before they are fully provisioned (i.e. fully yolked). In other words, by fertilizing eggs at earlier and earlier time points fertilization would eventually occur before eggs are fully provisioned and therefore the expression of the embryo genome would overlap the period during which the embryo is provisioned – and functional placentation would be achieved. This might be most likely to occur in lineages with intense sperm competition, where there is strong selection for males to be the first to fertilize available eggs. This hypothesis inverts the causation from female 'traits' (i.e. reproductive mode) driving patterns of male sexual selection to instead positing that male 'traits' (i.e. correlates of a high degree of sperm competition) govern the evolution of female reproductive mode. We now cite Haig [5] when this prediction is mentioned in the text.

3) What proportion of species diversity for poeciliinae has been sampled for this study, and how much overlap in sampling is there across variables? The authors account for incomplete species

sampling in their analyses of speciation rates, but I would like the authors to comment on the robustness of these results given the proportion of species sampled for this analysis.

There are 273 described species in the subfamily Poeciliinae (according to FishBase). The number of species with data for each character are as follows: courtship = 79 species, dichromatism = 94 species, ornamentation = 94 species, gonopodium length = 92 species, sexual dimorphism index standard length = 90 species, placenta = 150 species. This corresponds to between 29% and 55% of species in the subfamily sampled. These sample sizes can be found in Table S3. There was substantial overlap in sampling across variables. This can be visualized in Figure 3. All available data for male traits is displayed on the tips of the phylogeny in Figure 3, with blank cells corresponding to missing data. All species that had male data (i.e. all of those in Figure 3) also had data available on presence or absence of the placenta.

All species with which we had data were included in the analysis. In other words, there is not bias due to selective sampling or inclusion. Furthermore, the species from which data was gathered were not chosen based upon likely character state, so we believe it is likely to be a representative sample of character states in the subfamily. For these reasons we believe the analyses are likely to be reasonably robust.

References

1. Crespi, B. and C. Semeniuk, *Parent-offspring conflict in the evolution of vertebrate reproductive mode*. American Naturalist, 2004. **163**(5): p. 635-653.
2. Zeh, D.W. and J.A. Zeh, *Reproductive mode and speciation: the viviparity-driven conflict hypothesis*. Bioessays, 2000. **22**(10): p. 938-946.
3. Pollux, B.J.A., et al., *The evolution of the placenta drives a shift in sexual selection in livebearing fish*. Nature, 2014. **513**(7517): p. 233-236.
4. Zeh, J.A. and D.W. Zeh, *Toward a new sexual selection paradigm: Polyandry, conflict and incompatibility (Invited article)*. Ethology, 2003. **109**(12): p. 929-950.
5. Haig, D., *Sexual Selection: Placentation, Superfetation, and Coercive Copulation*. Current Biology, 2014. **24**(17): p. R805-R808.

Reviewers' Comments:

Reviewer #1:

Remarks to the Author:

The authors have suitably addressed all of the points that I raised, and I find the manuscript to be notably improved.

I believe this to be an excellent article that makes a novel and substantive contribution in an important area of research.

Reviewer #2:

Remarks to the Author:

I think the authors have done a great job revising their paper. My suggestions have been dealt with appropriately, and I have no further comments. I very much look forward to this important contribution in its published version.

Reviewer #3:

Remarks to the Author:

I applaud the authors' efforts to address my concerns about discretization of the Matrotrophy Index. The new analyses involving MI as a continuous trait are thorough and in good agreement with the results involving the binary trait. Fig. S20 is particularly convincing that inferences about placental origins are the same whether MI is binary or continuous. Overall, I found the presentation of this issue to enhance the apparent robustness of the results without bogging the manuscript down in methodological detail. The authors make reasonable and clear arguments for preferring to focus on the results involving MI as a binary trait.

In addition, I found the newly revised Introduction to be a marked improvement over the previous submission. The expanded rationale for hypothesized connections between maternal investment and sexual selection is especially clear and insightful.

David Collar